# Impact of Dairy Imports on Raw Milk Production Technology Progress in China

**DOI:** 10.3390/ijerph19052911

**Published:** 2022-03-02

**Authors:** Yuhang Bai, Li Li, Fengting Wang, Lizhong Zhang, Lichun Xiong

**Affiliations:** 1School of Economics and Management, Beijing Forestry University, Beijing 100083, China; yuhangbai2022@163.com; 2College of Economics and Management, Zhejiang Agriculture & Forestry University, Hangzhou 311300, China; lili621516@163.com; 3Zhejiang Province Key Cultivating Think Tank Research Academy for Rural Revitalization of Zhejiang Province, Zhejiang A&F University, Hangzhou 311300, China; 4Institute of Ecological Civilization, Zhejiang A&F University, Hangzhou 311300, China

**Keywords:** dairy products import, technological content, raw milk, production technological progress index, DEA

## Abstract

China’s dairy product import volume and output continue to grow rapidly, and to a certain extent, it will form a substitute for the Chinese dairy market. Therefore, it is necessary to study the impact of the import of dairy products on the technological progress of raw milk production in China. Using the data from 2005 to 2017, this paper uses the DEA model and the input-output model to analyze the impact of China’s dairy product imports on the technological progress of raw milk production. The model results show that: (1) there are differences in the technological content of dairy products from different importing countries; (2) The total technological content of imported dairy products hinders the improvement of the technological progress index of small, medium and large-scale production of raw milk in China, and has the most prominent negative impact on the technological progress of large-scale raw milk production in China; (3) The technological content of dairy products imports from New Zealand, Australia, Germany, the Netherlands and other countries can help improve the technological progress index of China’s moderate-scale production of raw milk, while importing countries from the United States, Canada and other countries hinder it.

## 1. Introduction

With the increase in per capita income and the popularization of healthy consumption concepts, Chinese consumers have a growing demand for dairy products [1]. The frequent occurrence of dairy product safety incidents has led consumers to pay more attention to the quality and safety of dairy products, and some consumers prefer imported dairy products with high quality and low prices [2]. Driven by the reform of the agricultural supply side, the development of China’s dairy industry is shifting from quantity-based to quality-based [3]. However, compared with developed countries in the international dairy industry, there is still a long way to go to improve the quality of milk, and the difference in resource endowments at home and abroad has led to a significantly higher price for domestic raw milk, even higher than the Cost, Insurance and Freight (CIF) of imported dairy products converted into raw milk. The result is that China’s dairy industry does not have a price advantage. From 1996 to 2017, China’s dairy product imports increased from 77,300 tons to 2,174,200 tons, a 28.13-fold increase, and the output of raw milk increased from 6,294,100 tons to 35,453,300 tons, a 5.63-fold increase. The growth rate of China’s dairy product imports far exceeds that of raw milk production [4]. The continued growth of imported dairy products is bound to squeeze the market share of domestic dairy products, intensify the competition in the domestic dairy market, and form a substitute for China’s raw milk market demand [5].

As the import trade of dairy products forms a substitute for China’s dairy market, it is worth pondering whether to promote the technological progress of China’s dairy industry. For importing countries, import trade is one of the main mechanisms of technology spillover [6,7]. The import trade of developing countries to developed countries is a process of introducing, absorbing, imitating, and re-innovating. International Research and Development (R&D) activities indirectly stimulate the technological progress of importing countries through trade in services or commodities [8,9,10]. Under normal circumstances, the technological content contained in imported goods is directly proportional to its quality level. Developing countries use “learning by doing” and other strategies to absorb advanced technologies contained in imported goods and accelerate the reduction of the technological gap between developing countries and developed countries [11]. New Zealand and Australia are among the developed regions of the world’s dairy industry, and their dairy farming technology, raw milk production efficiency, and dairy product processing technology occupy a leading position globally [12]. Jaforullah (1999) and MacDonald (2007) believe that the scale of dairy farming in New Zealand and the United States is gradually expanding, and that medium-scale and large-scale dairy farms will help reduce the cost of raw milk production, improve the overall production efficiency of the dairy industry, and increase operating income [13,14]. China’s dairy product imports have accelerated since 2008. According to data released by the Ministry of Agriculture and Rural Affairs, China’s large-scale dairy farms (referring to breeding scales of more than 100 cows) exceeded 50% for the first time in 2016, and large-scale dairy farming in 2018 amounted to 61.4%. In summary, large-scale farming is the development trend of the world’s dairy industry, and moderately large-scale operations are conducive to popularizing dairy farming technology, improving the technical efficiency of raw milk production, saving costs, and improving the competitiveness of the dairy industry [15]. In terms of technology spillover effects, China, as the world’s most important dairy product importer, is worth discussing in terms of whether its dairy product import trade with countries that are developed in the dairy industry will promote the technological progress of China’s raw milk production [16].

There are differences in the technological content of imported dairy products from different origins. Milk powder, cream, condensed milk, and other dairy products imported by China from New Zealand account for more than 80% of the total imports of similar dairy products [17,18]. Whey imports are mainly from the United States, Canada, the European Union, and other regions. Liquid dairy products are mainly imported from the European Union, with liquid dairy products from Germany and France accounting for about 47.81% of China’s total imports of liquid dairy products. Liquid dairy products imported from Australia and New Zealand account for nearly 50%. Dairy products such as cheese and butter are mainly from Denmark, Belgium, The Netherlands, New Zealand, and other countries. The origin of China’s dairy products has a diversified distribution. However, factors such as the provenance structure of dairy cows, breeding technology, mechanization of pastures, and natural resource endowments in various regions lead to differences in the quality and technological content of dairy products from different import sources [19,20,21,22]. Then, in the context of large-scale production of raw milk at home and abroad, what are the differences in the technological content of dairy products from different import source countries? What are the differences in the technological progress index of different large-scale producers of raw milk in China? Does the added value of technology contained in the import trade of dairy products promote or hinder the technological progress of China’s small-scale, medium-scale, and large-scale production of raw milk? What is the impact of dairy products from different import sources on the technological progress of China’s moderately large-scale raw milk production?

This paper focuses on the impact of the technological added value contained in dairy products from different import sources on the technological progress of raw milk production in China; analyze the impact of the surge in imports of dairy products on the production of raw milk in China from a broader perspective; and provide reference for improving the quality of China’s raw milk, the reputation of domestic dairy products, and the overall competitiveness of China’s dairy industry.

## 2. Literature Review 

Since the launch of China’s reform and opening up, the “Three-plus-one” trading mix has grown rapidly, and the country has actively undertaken international industrial transfer. After joining the WTO, China participated in international industrial division and cooperation with a more open attitude, but the added value of export commodities remains low. The 2008 economic crisis caused damage to China’s export trade, and the research focus shifted from the impact of export trade on economic development to import trade, believing that import structure and import quantity can promote economic growth [23,24,25]. The essence of economic development is a process of technical progress and industrial innovation, and the technological content of import trade has a significantly stronger effect on improving production efficiency than that of export trade [11]. In terms of dairy trade, existing studies on China’s dairy import trade mainly focus on analyzing the reasons for the surge in dairy imports [26]. In terms of raw milk production technology, Yu [27] believes that the raw milk production technology of Chinese family farms is significantly lower than that of large-scale farming. There is no study on the impact of dairy products from different import sources on the technological progress of raw milk production in China.

In economics, the technological content of products is usually expressed by the contribution of technology to the added value of products, and the technological content of the product indicates the level of quality. Technological added value is based on the machine equipment and manufacturing processes used in commodity production [28]. However, there are huge restrictions on the availability of technological input factors in different countries. Trade and income data of participating countries are usually used to measure the technological added value of traded products, and the quality of traded products is evaluated in terms of technological added value [29]. That is, the higher the technological added value of trade products, the higher their quality level [30,31]. In the import trade of dairy products, the technological added value of dairy products can include the raw milk quality, preservation technology, technological processes, and high-temperature sterilization technology used in the production process of dairy products, all of which reflect the technological content and technical level of dairy products in various countries [31]. In the process of assigning the technological content of dairy products with technological added value, the following theoretical basis should be followed.

The theory of comparative advantage holds that the difference in production technology between countries is the basis for the occurrence of international trade [32]. The exportable goods of a trading country depend on whether the product produced by the country has comparative advantage [33]. The higher the added value of the technology contained in the export product, the higher the wage level of the product produced by the country. It is believed that the technological added value of an export product can be expressed as the weighted average wage level of the country that exports the product divided by the country’s export value in the world’s total export value, where per capita Gross Domestic Product (GDP) can directly reflect the average wage of a country in most cases. Based on the comparative advantage theory and the “size of high and low level of wages reaction product technical added value” hypothesis, Sanjaya [34] analyzed the added value of products and suggested that the proportion of a country’s export volume in the country’s total export volume is the weight, and the technical added value of the country’s export products is expressed by the weighted average of the country’s per capita GDP. However, this method has the defect of overestimating the export capacity of large exporting countries. Fan Gang [28] supported the Heckscher-Ohlin (H-O) factor endowment theory. Hausmann [29] proposed the principle of value-added value assignment, and applied demonstrable comparative advantage to assign values to overcome the defect of overestimating the technological content of major exporting countries. However, this method takes the distribution of total export of products in each trading country as the weight to calculate, ignoring the difference between the distribution of product trade and the distribution of main production. This article combines the advantages and disadvantages of the above-mentioned methods and conducts empirical tests on the technological content of China and its trading partners or trade in services to further confirm the feasibility of the above-mentioned methods.

## 3. Model Setting and Data Sources

### 3.1. Measurement of Technological Added Value

In this section, the technological content index of China’s imported dairy products is constructed based on the above calculation method of technical added value of trade products. Based on the theory of cost comparative advantage, the weight was taken as the revealed comparative advantage of dairy products from different import countries, and according to the following hypothesis, dairy products produced in high-income countries have high technology content, while dairy products produced in low-income countries have low technology content [35]. Therefore, there is a positive relationship between raw milk production technology level and per capita GDP, and the per capita GDP index is helpful to compare the development level of dairy productivity in major dairy-producing countries. Based on the above logic, this paper intends to calculate the technological content of dairy products from each imported source country. First, the explicit comparative advantage of dairy products from each imported source country is calculated, as shown in Formula (1):(1)RCA idairy=XidairyXiXwdairyXi
where $X_{i}^{dairy}$ is the export value of different types of dairy products in the country *i*, and $X_{w}^{dairy}$ represents the export value of different kinds of dairy products in the world. $X_{i}$ and $W_{i}$ are the total exports of the country i and the world, respectively. Secondly, the weight of indicative comparative advantage of export dairy products of import source countries after standardized treatment can be expressed as:(2)Widairy=RCAidairy∑i=1nRCAidairy

The technological added value of imported dairy products, which represents the technological content of imported dairy products, can be obtained:(3)$$TC^{dairy}=\sum_{i=1}^{n}W^{dairy}Y_{i}$$

$Y_{i}$ is country i’s per capita GDP calculated at purchasing power parity (PPP). The technological level of dairy products imported by China from the country i can be expressed as follows:(4)$$ETC=\sum_{i=1}^{m}TC^{dairy}\times \frac{X_{ci}^{dairy}}{X_{i}^{dairy}}$$
where m is imported dairy products, including liquid milk, milk powder, yogurt, whey, butter, cheese, and six other categories of dairy products. $X_{ci}^{dairy}$ is the total amount of dairy products imported by China from country i, and $X_{i}^{dairy}$ is the total export volume of dairy products from country i.

### 3.2. Malmquist TFP Index Model

Regarding the technological progress of raw milk production, this section first uses the Malmquist index method based on Data Envelopment Analysis (DEA) to decompose the Malmquist total factor productivity (TFP) of the large-scale production of raw milk in China, and then obtain the technological progress. The technological progress index describes the movement of the production technology boundary, which can be expressed as technological progress. The Malmquist index method was first proposed by the Swedish economist Malmquist; Caves and Diewert (1980) first applied it to the study of production efficiency, but the method to measure the distance function was not available until the emergence of the DEA method [36], that is, through the Linear planning method to measure technological efficiency. In order to facilitate the observation of total factor productivity, it is decomposed into comprehensive efficiency and technological progress. The Malmquist index measuring TFP from period t to period t+1 is defined as:(5)$$M_{0}\left(x_{t+1},y_{t+1},x_{t},y_{t}\right)={\left[\frac{d_{0}^{t}\left(x_{t+1},y_{t+1}\right)}{d_{0}^{t+1}\left(x_{t+1},y_{t+1}\right)}\times \frac{d_{0}^{t}(x_{t+1},y_{t+1)}}{d_{0}^{t+1}\left(x_{t},y_{t}\right)}\right]}^{\frac{1}{2}}$$
where ($x_{t},y_{t}$) and ($x_{t+1},y_{t+1}$) represent the input and output vectors in period t and t+1, respectively; $d_{0}^{t}$ and $d_{0}^{t+1}$ represent the distance function of the period *t* and t+1, respectively, with the technology $E^{t}$ as the reference in the t period.

Assuming that the return to scale is constant, the Malmquist index can be decomposed into the technical efficiency change index (*effch*) and the technological progress index (*techch*). The method is as follows:(6)$$M_{0}\left(x_{t+1},y_{t+1},x_{t},y_{t}\right)=\frac{d_{0}^{t+1}\left(x_{t+1},y_{t+1}\right)}{d_{0}^{t}\left(x_{t},y_{t}\right)}\times {\left[\frac{d_{0}^{t}\left(x_{t+1}.y_{t+1}\right)}{d_{0}^{t+1}\left(x_{t+1},y_{t+1}\right)}\times \frac{d_{0}^{t}\left(x_{t+1},y_{t+1}\right)}{d_{0}^{t+1}\left(x_{t},y_{t}\right)}\right]}^{\frac{1}{2}}=effch\times techch$$

When the restriction conditions are added, assuming that the return to scale is variable, the distance function of each decision-making unit is solved, and the technical efficiency change index (*effch*) is decomposed into the scale efficiency index (*sech*) and the pure technical efficiency index (*pech*). Therefore, total factor productivity can also be expressed as:(7)$$M_{0}\left(x_{t+1},y_{t+1},x_{t},y_{t}\right)=techch\times sech\times pech$$

In summary, Malmquist index is the product of pure technical efficiency index, scale efficiency index and technological progress index. Pure technical efficiency is used to measure changes in the promotion and use of existing technologies and varieties; scale efficiency is used to measure changes in the scale of production and management, organization, and management; and technological progress is used to measure the development of new technologies and varieties. There are three changes in the TFP index. When TFP = 1, there is no change in total factor productivity; when TFP > 1, total factor productivity has improved; when TFP < 1, total factor productivity has deteriorated.

For the measurement of the raw milk production technology progress index, the selected input and output indicators are all indicators commonly used by scholars in the past to measure China’s raw milk production technology index, and the factor input indicator is labor input ($X_{1}$, including family working days and employment days; working days), feed costs ($X_{2}$, including concentrated feed and green roughage costs; yuan), fixed asset input ($X_{3}$, including feed processing costs, tool and material costs, repair and maintenance costs, fixed asset depreciation; yuan), and other expenses ($X_{4}$, including water, fuel and power costs, medical expenses; yuan). The output indicator is raw milk production ($X_{5}$, kg). This section uses the DEAP2.1 software to measure the technological progress index. A numerical result greater than 1 indicates that the technological progress is growing positively, and less than 1 indicates the opposite.

Guided by the idea of the technology gap model, based on the new trade theory and learning from the empirical research of Fu (2005), the model is constructed to mainly examine the technological content represented by the added value of dairy products in different import source countries, which is a key variable for domestic raw milk [37]. At the same time, control variables are set according to consumer demand and the cost/profit ratio. After the “Sanlu” incident in 2008, China’s dairy industry was hit hard, and the Chinese government has since been working to resume dairy production. The government introduced support policies for the dairy industry. The dairy farming subsidy policy takes large-scale farms as the main subsidy object (The dairy farming subsidy policy comes from *China Dairy Industry Statistical Yearbook* (2018)), and dairy industry policy is used as a dummy variable. In summary, this article intends to introduce the consumption level of dairy products of urban and rural residents, the profit rate of raw milk production costs as control variables, and the year when China promulgated the milk industry policy as dummy variables, and then study the impact of dairy product import technology content on the production of medium- and large-scale raw milk. The impact model of imported dairy products’ technological content is as follows:(8)$$techch_{it}=\alpha_{0}+\alpha_{1}etc_{it}+\alpha_{2}city_{it}+\alpha_{3}rural_{it}+\alpha_{4}profit_{it}+\alpha_{5}policy_{it}+w_{it}$$
where *i* is the country; *t* is the year; *techch* is China’s raw milk production technology progress index measured by the Malmquist index method, representing the quality of China’s raw milk; *etc* is the technological added value of imported dairy products from the main import source countries of dairy products, indicating the level of technological content; *city* represents the consumption expenditure of urban residents and the proportion of disposable income; *rural* represents the proportion of rural residents’ consumption expenditure to disposable income; *profit* represents the profit rate of raw milk production costs; *policy* represents dairy industry policy; and *w* is the residual item.

### 3.3. Data Sources

Part of this section uses UNcomtrade database from 2005 to 2017 to determine the types of dairy products imported according to the standards of the customs code HS1996, mainly including liquid milk (0401), milk powder (0402), yogurt (0403), whey (0404), butter (0405), cheese (0406), and six other categories. To be designated as a main source of certain dairy product imports, must account for more than 5% of China’s total imports of this category of dairy products for two consecutive years. Based on the data, the main sources of China’s dairy product imports include New Zealand, Australia, the United States, Canada, Uruguay, France, Germany, the United Kingdom, Denmark, The Netherlands, Finland, Belgium, Ireland, and Poland; China’s imports of various dairy products, the export of dairy products of the 14 countries mentioned above, the total export trade volume of a particular country, and the total amount of dairy products imported by China from this country all come from the Uncomtrade database. The importing countries’ per capita GDP are assessed according to purchasing power parity, and the data come from the World Bank database. The input–output data used in the domestic raw milk production technology progress index—namely, concentrated feed, green roughage, feed processing costs, tool and material costs, repair and maintenance costs, depreciation of fixed assets, water costs, fuel power expenses, medical and epidemic prevention expenses, and raw milk production—are all from the National Agricultural Product Cost and Benefit Data Collection (2006–2018), which uses small-scale, medium-scale, and large-scale raw milk input and output in 27 major raw milk-producing provinces across the country. As a sample, the price index of agricultural production materials based on 2004 is used to deflate the price in the input–output data. The control variables of urban and rural residents’ consumption expenditure and their disposable income are from the *China Statistical Yearbook* (2006–2018). Raw milk production cost and profit margin come from the Compilation of National Agricultural Product Cost and Benefit Data (2006–2018). Dairy industry policy as a dummy variable is mainly collected from the websites of the State Council, the Ministry of Agriculture and Rural Affairs, and the *China Dairy Industry Yearbook* over the years. After the discovery of the “melamine” tainted milk powder incident in 2008, the Chinese government intensively promulgated milk industry support policies in order to resume dairy production. Based on this, the milk industry policy for 2005–2008 was set to 0; the milk industry policy for 2009–2017 was set as 1.

## 4. Empirical Result Analysis

### 4.1. Results of the Imported Technology Content of China’s Dairy Products

According to the above steps, the technological content of China’s dairy products is calculated, imported from 14 countries including New Zealand, Australia, the United States, Uruguay, France, and Germany. As analyzed above, the technological content of imported dairy products is expressed by the added value of technology. The level of technological content can reflect the quality of dairy products, and explore whether imported high-quality dairy products can improve the technical level of raw milk production in China and improve the quality of raw milk.

The technological content of dairy products exported to China varies significantly among the importing countries of dairy products. As shown in Table 1, dairy products imported from New Zealand have the highest technological content, rising from 746.21 in 2005 to 6284.67 in 2017, and reaching a peak of 7344.33 in 2014, with periodic fluctuation characteristics. Although the technological content of dairy products imported from Australia by China is much lower than that of New Zealand, the technological content of dairy products still ranks second among China’s main import sources. The technological content of dairy products imported from Australia by China increased from 42.62 in 2005 to 173.53 in 2017. The technological content of dairy products imported from Uruguay is much higher than that of the United States and Europe, increasing from 0.88 in 2005 to 56.49 in 2017 and reaching a peak of 352.14 in 2014. However, the technological content of dairy products exported from Uruguay to China dropped significantly after 2014. In North America, the technological content of dairy products imported from the U.S. by China is much higher than that of dairy products imported from Canada. The technological content of dairy products imported from the U.S. increased from 8.4 in 2005 to 35.98 in 2017, while the technological content of dairy products imported from Canada decreased from 1.49 in 2005 to 0.25 in 2017. Dairy products imported from European countries have relatively close technical levels. The technological content of dairy products imported from France ranks first among European countries, rising from 9.69 in 2005 to 75.68 in 2017. The technological content of dairy products imported from Ireland, Denmark, and Finland is relatively close. From 2005 to 2017, the technological content of China’s dairy products imported from Ireland rose from 14.5 to 61.95, the technological content of dairy products imported from Denmark rose rapidly from 2.97 to 58.81, and the technological content of dairy products imported from Finland rose from 18.4 to 57.61. However, the technological content of dairy products imported from The Netherlands, Germany, Poland, the United Kingdom, and Belgium increased slightly. From 2005 to 2017, the technological content of dairy products imported from The Netherlands increased from 4.95 to 33.97, the technological content of dairy products imported from Germany increased from 0.38 to 23.95, and the technological content of dairy products imported from Poland increased from 1.44 to 16.06. The technological content of dairy products exported to China from the U.K. and Belgium has been the slowest to increase.

The above analysis further proves that the technological contents of dairy products from different source countries are very different. This prompts the question, what impact will the technological content of imported dairy products have on the quality of raw milk in China, that is, on the technological progress of large-scale production of raw milk?

### 4.2. Result of the Technology Progress Index of China’s Large-Scale Raw Milk Production

In this section, DEA2.1 software is used to measure the technological progress index of small-scale, medium-scale, and large-scale raw milk production in 27 provinces in China using input-oriented methods. The 27 provinces include Hebei, Shandong, Henan, Liaoning, Inner Mongolia, Heilongjiang, Beijing, Tianjin, Shanghai, Chongqing, Qinghai, Xinjiang, Gansu, and Mingxia. Specific results are shown in Table 2.

The technological progress of China’s raw milk production has obvious segmental characteristics. In terms of stages, the index of technological progress for small-scale, medium-scale, and large-scale raw milk production from 2005 to 2007 is in a decreasing range, of which the large-scale raw milk production technological progress index declined the most, with a decrease of 62%, followed by small-scale production, with a decrease of 52.68%. The technological progress index of medium-scale raw milk production has a relatively small decrease of 43.63%. In 2008, the raw milk technology progress index under the three models rebounded rapidly. From 2009 to 2014, China’s raw milk production technology progress index fluctuated frequently. However, from 2015 to 2017, the technological progress of small-scale raw milk production maintained a slow growth. In 2017, the technical progress indexes of medium-scale and large-scale raw milk production were 0.937 and 0.69, respectively, which were 31.1% and 58.86% lower than in 2015.

The reason why China’s raw milk technology progress index fluctuates frequently around 2008 may be due to the significant increase in dairy farming costs in 2007. The purchase price of raw milk was low, 40% of the country’s dairy farms suffered serious losses, and cows were slaughtered in some areas, threatening the foundation of China’s dairy industry development. In 2008, the raw milk production technology progress index increased significantly compared with 2007. However, the 2008 melamine incident severely disrupted China’s normal raw milk production. At the same time, the international financial crisis caused the prices of bulk commodities to soar and plummet. The Chinese government introduced a series of milk industry support policies to resume raw milk production and strengthen the supervision of raw milk production. That is to say, since 2009, the impact of dairy industry policies and economic recovery on the technological progress of raw milk has gradually appeared, and has remained above 1 throughout the years. From 2015 to 2017, China’s large-scale raw milk production technology progress index had been significantly lowered, and the reason may be related to environmental protection policies. The cost of environmental protection transformation of large-scale dairy farms is too high, which has led to a significant increase in the production cost of raw milk. However, the increases in raw milk production and in raw milk purchase prices have been relatively slow, so the technology progress index of large-scale raw milk production has declined significantly.

China’s small-scale raw milk production technology progress index is relatively high. By comparing the mean value of the technology progress index of small-scale, medium-scale, and large-scale raw milk producers, it is found that the mean value of small-scale raw milk production technology progress index is 1.050, which is slightly higher than the mean value of medium-scale’s 1.030 and mean value of large-scale’s 1.027. There are several possible reasons for this difference. First, modular production is a cyclical and gradual process. If the progress from free-range to small-scale, from small-scale to medium-scale, and from medium-scale to large-scale is too fast, this will lead to higher debt ratios for raw milk producers, increased capital costs, and higher depreciation costs of fixed assets for large-scale production. Second, in the process of advancing from small-scale to large-scale, the infrastructure and medical and health conditions for raw milk production have not been improved simultaneously, which may increase the probability of epidemics and increase the unit cost of raw milk production. Third, the large-scale production of raw milk is not conducive to the localization and low cost of forage, and it is also not conducive to the realization of integrated planting and breeding production methods. At the same time, it increases the difficulty of manure treatment, resulting in a significant increase in the comprehensive production cost of medium-scale and large-scale raw milk.

### 4.3. Analysis of the Influence of the Total Imported Technology Content of Dairy Products on the Technological Progress of China’s Raw Milk Production

On the basis of measuring the technological content of imported dairy products and the technology progress index of the large-scale production of raw milk. Stata software was used to analyze the total content of imported technical of dairy products from 14 countries, including New Zealand, Australia, the United States, and Germany, which will affect China’s small-, medium-, and large-scale raw milk production technology progress index. In view of the excessive imported technological content of dairy products by some countries, it is necessary to logarithmize the technological content of dairy products in the 14 import source countries and technological contentderive the total technological content of China’s imported dairy products, and use this as the variable value of the technological content of imported dairy products. From performing the Hausmann [29] test on the regression model, it was found that the results of the random effects processing regression model are better than the fixed effects model. Therefore, the random effects model is selected to regress the panel data. The model estimation results are shown in Table 3.

The total imported technology content of dairy products has a negative impact on the technological progress of China’s medium-scale raw milk production. According to the regression results, the total imported technology content of dairy products has a negative impact on the technology progress index of China’s small-scale raw milk production at a significant level of 10%. However, the impact is small, only −0.005. That is, every 1% increase in the total content of imported dairy products will result in a 0.5% decrease in the technological progress of China’s raw milk production. The control variables such as the consumption level of urban residents, the consumption level of rural residents, the cost to profit ratio, and dairy industry policy have no significant impact on the technological progress of China’s raw milk production.

The total imported technology content of dairy products has a negative impact on the technological progress of China’s medium-scale raw milk production. It can be seen from Table 3 that the total imported technology content of dairy products has a significant negative impact on China’s medium-scale raw milk production technology progress index at a significance level of 1%. The coefficient is −0.011, that is, every 1% increase in the total imported technology content of dairy products will result in a 1.1% decline in the technological progress of China’s medium-scale raw milk production. The consumption level of urban residents has no significant impact on the technology progress index of medium-scale raw milk production. At the 5% significance level, the consumption level of rural residents has a positive impact on the technological progress of China’s raw milk. At the 10% significance level, the raw milk cost and profit margin have a positive effect on the technological progress of China’s raw milk production. At the 1% significance level, dairy policy has significantly stimulated the positive growth of China’s raw milk production technology progress index.

The total imported technology content of dairy products also has a negative impact on the technology progress index of China’s large-scale raw milk production. The total imported technology content of dairy products at the 1% significance level has a negative impact on China’s raw milk production technology, with a coefficient of −0.016. That is, for every 1% increase in the technological content of imported dairy products, China’s raw milk mass production technology progress index will drop by 1.6%. The consumption levels of urban residents and rural residents have no significant impact on the technology progress index of China’s large-scale raw milk production. At the 1% significance level, dairy industry policy has a positive impact on the technological progress of China’s raw milk.

The total imported technology content of dairy products has a significant negative impact on China’s raw milk production technology progress index. The total content of imported dairy products has a negative impact on the technological progress of China’s small-, medium-, and large-scale raw milk production, which is contrary to expectations. The reason may be that China’s dairy imports are mainly dry dairy products, while domestic dairy production is dominated by liquid dairy products. In 2017, China imported a total of 2,470,500 tons of dairy products, of which 1,768,800 tons were dry dairy products, accounting for approximately 71.6% of the total imports (China’s 2017 dairy import data comes from this website: https://www.chyxx.com/industry/201807/656247.html, accessed it was 26 January 2022). It can be seen that the technological added value contained in imported dairy products is mainly reflected in the processing of dry dairy products. The technological content of imported dairy products can help improve China’s dry dairy production technology. Dairy companies pursue profit maximization, thereby reducing investment in liquid dairy production. The production technology of domestic liquid dairy products has slowly improved, and even negative growth has occurred. China’s raw milk production in 2017 was 35.45 million tons, and the raw milk consumed in the production of domestic liquid dairy products accounts for 75.93% of the total raw milk output, indicating that China’s raw milk production is mainly used for other liquid dairy products. The liquid dairy production and raw milk production technology level change directions are more consistent. To sum up, China’s dairy trade and production is structure dominated by dry dairy products, and dairy products dominated by liquid dairy products are mismatched. As a result, China’s raw milk production technology progress index has been slow to increase.

The total imported technology content of dairy products has a stronger negative impact on the technological progress of large-scale raw milk production than that of medium-scale and small-scale production. Horizontal comparison indicates that the large-scale production technology progress index of raw milk is facing the most prominent negative impact, followed by medium-scale production. The small-scale raw milk production technology progress index is relatively less affected. The reason is that the large-scale production of raw milk refers to the production model in which the number of dairy cows is more than 500, and the investment in infrastructure and mechanization is much higher than that of small- and medium-sized producers. During the production of raw milk, large-scale production is linked to higher forage consumption, difficulty in disease prevention, higher fuel costs and labor costs, and fixed asset depreciation. According to the above analysis, it can be seen that China’s raw milk production is mainly used to produce liquid dairy products. The imported technology content of dairy products hinders the technological progress of domestically produced liquid dairy products, and thus the technological progress of China’s raw milk production. Therefore, the technological content of imported dairy products hinders the progress of large-scale production of raw milk more than the small- and medium-scale production modes of raw milk.

Dairy industry policy as a dummy variable has a significant positive role in promoting the technological progress of medium-scale and large-scale raw milk production. However, the promotion of the small-scale raw milk production technology progress index is not significant. The main reason is that China’s dairy industry subsidy policy is oriented towards large-scale farming. In 2011, the subsidy standard for standardized scale breeding of dairy cows was raised from 200 to 300 in stock. In 2017, the central financial subsidies granted 800,000, 1.3 million, and 1.7 million subsidies to farms with 300–499 heads, 500–999 heads, and 1000 heads or more, respectively. Subsidies for small-scale (11–50 heads) production of raw milk essentially do not exist. To a certain extent, financial subsidies can promote large-scale (over 500) and medium-scale (51–500) dairy farms to introduce high-quality dairy cows, improve management levels, improve medical and epidemic prevention systems, etc. Therefore, financial subsidies have a significant positive stimulus effect on the large-scale and medium-scale production technology progress indexes of raw milk. At the same time, this indirectly proves that dairy industry policy has played a positive role in the development of China’s dairy industry.

### 4.4. The Impact of the Total Imported Technology Content of Dairy Products on the Technological Progress of Moderately Large-Scale Raw Milk Production

The above analysis postulates that the total technological content of imported dairy products has a significant negative impact on the production technology progress index of China’s small-scale, medium-scale, and large-scale of raw milk. However, the differences in the type, structure, and technological content of dairy products from different import sources must be considered. In countries such as New Zealand and the United States, the dairy products exported to China are mainly dry dairy products. In Germany, Australia, and other countries, the dairy products exported to China are mainly liquid dairy products. This leads us to ask, will the technological content of the dairy products exported to China from various import source countries have a negative impact on the moderately large-scale production of raw milk in China?

Appropriate scale production of raw milk is the future development direction of China’s dairy industry. Many researchers agree that the moderately large-scale production of raw milk is helpful to prevent excessive large-scale production of raw milk from causing excessive cost rises, high demands for forage, and dairy cattle farming exceeding the environmental carrying capacity, as well as to prevent an increase management, sanitation, and epidemic prevention costs and manure pollution. It is believed that the cost of large-scale raw milk production in China is higher than that of small-scale and medium-scale production. Looking at other countries, the production scale of raw milk in New Zealand is dominated by farms with 100–499 heads of dairy cattle; the United States is dominated by more than 1000 heads; and The Netherlands is mainly 100–149 heads. The proportion of a production scale of more than 150 heads has a tendency to increase faster [38]. In summary, each region should operate based the local geographic environment, related industries, and resource endowments to ensure a reasonable choice of production scale [39,40]. According to the calculations of the present work, the average cost and profit rate of raw milk production of different scales in China from 2005 to 2017 are as follows: small-scale is 32.16%; medium-scale is 24.81%; and large-scale is 23.01%. That is, with the expansion of the production scale of raw milk, the cost and profit rate of raw milk has gradually decreased, resulting in diseconomies of scale. In 2016, the proportion of large-scale farms of more than 1000 dairy cows in China exceeded 50%, according to the deployment of the Ministry of Agriculture’s dairy farming plan. During the 13th Five-Year Plan period, the proportion of large-scale dairy farms exceeded 70% in China. Altogether, it is necessary to consider China’s raw milk production resource endowment conditions, the changes in the profit margin of large-scale production costs, the development plan of the dairy industry, and the disadvantages of large-scale breeding. At the same time, it is important to learn from the experience of large-scale raw milk production in countries such as New Zealand. We believe it is reasonable to consider the medium-scale production of raw milk (51–500 heads) as the development direction of China’s dairy industry on a moderate scale. Then, it is possible determine the impact of the technological content of dairy products exported to China from different countries on the technological progress of China’s moderately large-scale raw milk production.

In this work, regression was performed with Stata software, and according to the results of Hausman’s test, a random effects model was selected for parameter estimation. The regression results from the model are shown in Table 4. At the 1% significance level, the technological content of dairy products imported by China from New Zealand, Australia, Germany, and The Netherlands has a significant positive impact on the technological progress of China’s moderately large-scale raw milk production. The technological content of New Zealand dairy products has the greatest promotion effect on the technological progress of China’s raw milk production. That is, every change in the technological content of New Zealand dairy products by one unit will promote an increase of 1.25 units in the technical progress index of the moderately large-scale production of raw milk in China. Next is The Netherlands: every change in the technological content of its dairy products by one unit will drive China’s moderately large-scale production of raw milk technology progress index to increase by 0.94 units. The technological contents contained in dairy products in Germany and Australia have a relatively small positive effect on the technological progress index of moderately large-scale raw milk production in China. The United States, Canada, the United Kingdom, Denmark, Finland, and Belgium have a significant negative impact on the technological progress of China’s moderately large-scale raw milk production. In particular, the technological content of imported dairy products from the United States, Belgium, and Denmark pose a relatively large hindrance to the technical progress index of China’s moderately large-scale raw milk production, while the obstacles in the U.K. and Canada are relatively small. As for the technological content of dairy product imports from France, Ireland, Poland, and other countries, the impact on the technological progress of China’s moderately large-scale raw milk production is missing. The reason may be that the production technology, processing equipment, and management level of dairy production in some developed countries are relatively close or consistent, which led to the omission of the technological content of dairy products imported from these countries due to multicollinearity.

There are three specific reasons why the technological contents of dairy products imported from New Zealand, Australia, Germany, and The Netherlands have a positive impact on China. First, liquid dairy products imported from New Zealand, Australia, Germany, and other countries account for about 80% of China’s total imports of liquid dairy products, and 75.92% of raw milk in China is used to produce liquid dairy products. Therefore, the technological contents of liquid dairy products in New Zealand, Australia, and Germany have a significant positive spillover effect on the technological progress of China’s moderately large-scale raw milk production. Second, China signed free trade agreements with New Zealand and Australia and therefore, they have a closer relationship in dairy trade and dairy production technology exchanges [12]. The third reason is evidenced by reports in the China Dairy Industry Yearbook about Chinese dairy companies “going to sea” after the “melamine” incident, which put Chinese dairy companies in fierce competition with overseas milk sources. In 2009, Yashili signed a pasture farming agreement with New Zealand. In 2010, Bright Dairy acquired 51% of New Zealand’s New Wright. In 2012, Ausnutria acquired the entire equity of Hyproca. In 2014, Yili invested USD 2 billion in New Zealand to establish a milk source base, and Beinmate established a subsidiary in Ireland. In 2015, Beinmate acquired a 51% stake in Fonterra’s Daren plant in Australia for USD 367 million. In 2017, Ausnutria acquired 100% stake in Australia ADP and 50% stake in Ozfarm. In 2018, Yili acquired Westland, New Zealand’s second largest dairy cooperative. From the above-mentioned overseas investment behaviors of Chinese dairy companies, it can be seen that New Zealand and Australia are the first choices for investment by Chinese dairy companies. Dairy companies building dairy farms and dairy processing plants in the above-mentioned countries is conducive to learning the raw milk production technology, management experience, and dairy processing technology of these countries, leading to the rapid growth of China’s raw milk production technology progress index. In summary, factors such as dairy product import structure, China’s dairy product production structure, overseas investment behavior of dairy companies, and free trade agreements which determine the technological content of dairy products imported from New Zealand, Australia, Germany, The Netherlands and other countries can promote the improvement of the technology progress index of moderately large-scale raw milk production in China.

The United States, Canada, Finland, and other countries have hindered the progress of China’s raw milk technology. The reason may be that the dairy products China imports from the United States are mainly whey, accounting for 54.9% of the total imports. As China itself produces little whey, this may affect China’s ability to absorb and benefit from the technological content of these American dry dairy products. Therefore, the technological content of dairy products imported from the United States has a negative impact on the technological progress of China’s raw milk production. The reasons for the negative impact of the technological content of dairy products from Canada, Denmark, the United Kingdom, Finland, and Belgium on raw milk production in China are similar to the United States—the export structure of these countries’ dairy products is inconsistent with the structure of China’s dairy production.

## 5. Conclusions

In this paper, Malmquist TFP Index Model was used to decompose the technical progress index of China’s raw milk scale production through input-output. Based on the theory of comparative advantage, the index of dairy product display comparative advantage was obtained, and then the technological content of Chinese dairy product import was calculated; the stochastic effect model was established to empirically analyze the influence of technological content of dairy imports on technological progress index of raw milk scale production from 2005 to 2017, and the following conclusions were drawn:
(1)There are significant differences in the technological content of dairy products imported from the main source countries. The technological content of dairy products imported by China from New Zealand is always higher, followed by Australia. The technological content of dairy products imported from the United States is obviously higher than that of Canada; Among European countries, the technological content of dairy products imported from France is the highest, while that from Ireland, Denmark and Finland is relatively close, while that from Netherlands, Germany, Poland and the United Kingdom has a small increase.(2)The technological progress index of scale production of raw milk in China has stage characteristics. From 2005 to 2007, the technological progress index of large-scale production of raw milk decreased obviously, in which the technological progress index of large-scale production of raw milk decreased most obviously, followed by medium-scale production, while the small-scale raw milk production technological progress index decreased least. The production technological progress index of scale raw milk in China fluctuated frequently from 2009 to 2014. From 2015 to 2017, only small-scale raw milk production technology progress index maintained a small increase, while medium-scale and large-scale raw milk production technology progress index once showed a decline of more than 30%.(3)The total import technology content of dairy products has a negative impact on the technological progress index of different scale production of raw milk in China, and the negative impact on the large-scale production of raw milk is the most serious. The import structure of Chinese dairy products is contrary to the production structure. Namely, the production of Chinese dairy products is mainly liquid dairy products, while the import is mainly dry dairy products, so the technological content of Chinese dairy products import is mainly reflected in the technological content of dry dairy products, rather than liquid dairy products. However, China’s dry dairy production is about one 7 of the liquid dairy products production, low capacity to a certain extent, shows the product production technology level is not high, so China’s existing dry dairy production technology hinder the absorption of China’s dairy industry to import dairy products technology content, and negative influence on China’s scale production of raw milk. As a dummy variable, dairy policy has a significant positive effect on the technological progress index of medium-scale and large-scale raw milk production in China, which is mainly because China’s dairy industry policy subsidies focus on medium and large-scale dairy farms.(4)According to the country classification, the impact of imported dairy product technology content on China’s moderate scale production technology progress index of raw milk is calculated, and there are significant differences in the results. Previous researches suggest that moderate scale production of raw milk is China’s dairy industry development direction in the future, based on medium-scale of raw milk production to represent the moderate scale in the article, the results showed that dairy technology content on the moderate scale of production and technology of China imported from New Zealand, Australia, Germany, the Netherlands and other countries has a significant positive role in promoting progress index, However, the technological content of dairy products imported from the United States, Canada, The United Kingdom, Denmark, Finland, Belgium and other countries has hindered the growth of China’s technological progress of raw milk. The main reason for this difference is a combination of factors, such as the type and structure of dairy products exported to China by various countries, the production structure of China’s dairy products, the overseas investment behavior of dairy companies, and trade agreements.

Furthermore, As we all know that China has a large population, and its per capita arable land resources and per capita water resources are lower than the world average. At the same time, it does not have comparative advantages in agricultural breeding, crop cultivation management, cost control, and large-scale production. Therefore, China needs to import a large amount of agricultural product that with lower price and higher quality every year to meet domestic demand. In actual international trade, China imports a large amount of soybeans, corn, wool and other agricultural products every year, so the research can be extended to the impact of the technological content of soybeans, corn, wool and other agricultural products on the technological progress of related agricultural products in China.

## Figures and Tables

**Table 1 ijerph-19-02911-t001:** The technical content of dairy products in China’s major import sources of dairy products.

**Time**	**New Zealand**	**Australia**	**USA**	**Canada**	**Uruguay**	**France**	**Germany**
2005	746.205	42.615	8.404	1.486	0.878	9.693	0.384
2006	966.084	41.245	10.845	1.198	1.747	11.679	0.701
2007	812.595	54.112	11.242	1.554	14.668	17.669	1.323
2008	787.765	63.553	14.147	1.713	5.389	17.502	1.259
2009	1893.989	49.729	12.011	2.463	2.569	20.339	2.399
2010	3014.137	55.249	15.797	0.972	53.256	15.563	3.355
2011	2474.059	45.861	21.588	0.289	46.976	19.768	5.719
2012	3961.462	51.385	23.091	0.226	50.918	30.385	9.796
2013	5993.624	82.574	36.535	0.093	287.057	36.494	12.613
2014	7344.327	122.051	39.478	0.359	352.136	48.299	17.177
2015	4360.186	151.757	26.282	0.172	105.162	43.131	19.321
2016	5061.959	170.896	25.277	0.309	135.981	58.940	21.283
2017	6284.666	173.533	35.981	0.248	56.494	75.676	23.946
**Time**	**UK**	**Denmark**	**The Netherlands**	**Finland**	**Belgium**	**Ireland**	**Poland**
2005	0.266	2.968	4.948	18.399	1.347	14.497	1.441
2006	0.107	4.541	4.459	18.069	0.645	13.754	2.809
2007	0.059	4.858	6.574	27.647	0.566	16.633	1.576
2008	0.015	3.271	7.552	18.886	1.067	17.233	0.452
2009	0.147	10.526	9.500	35.116	1.009	20.736	1.923
2010	1.046	38.035	8.190	31.913	3.177	20.800	3.595
2011	1.648	28.609	12.901	37.326	1.921	34.372	5.867
2012	1.344	23.355	14.324	63.680	3.559	39.148	5.116
2013	2.094	28.733	19.342	64.666	4.531	51.784	8.051
2014	4.725	44.311	18.281	73.860	5.631	55.643	15.755
2015	3.885	45.138	22.762	52.941	4.079	62.324	14.531
2016	6.198	43.538	25.629	52.934	5.511	52.144	12.440
2017	8.635	58.811	33.971	57.605	7.064	61.948	16.059

Data source: Uncomtrade website.

**Table 2 ijerph-19-02911-t002:** China’s raw milk production technology progress index from 2005 to 2017.

Time	Small Scale (10–50 Cows)	Medium Scale (50–500 Cows)	Large Scale (500+ Cows)
2005	1.026	0.956	1.043
2006	1.697	1.201	1.400
2007	0.803	0.677	0.532
2008	1.068	1.213	1.183
2009	1.013	1.208	1.125
2010	1.221	1.003	0.626
2011	0.829	1.069	1.335
2012	0.800	0.768	1.109
2013	1.408	1.182	0.960
2014	0.803	0.642	0.860
2015	1.011	1.360	1.677
2016	0.941	1.180	0.809
2017	1.035	0.937	0.690
Mean Value	1.050	1.030	1.027

Note: China’s raw milk production technology progress index is calculated according to the “National Agricultural Product Cost and Benefit Data Collection (2006–2018)”; The number of dairy cows determines the scale of raw milk production farms, Chinese scholars, Statistical yearbooks usually define 10–50 cows as small-scale raw milk production farms; 50–500 cows as medium-scale raw milk production farms; 500+ cows as large-scale raw milk production farm.

**Table 3 ijerph-19-02911-t003:** The influence of imported technology total content of dairy products on the technological progress index of the large-scale production of raw milk.

Variable	Small-Scale Technological Progress Index		Medium-Scale Technological Progress Index		Large-Scale Technological Progress Index	
	Coef	SE	Coef	SE	Coef	SE
etc	−0.005 *	0.003	−0.011 ***	0.003	−0.016 ***	0.004
city	−0.001	0.004	0.002	0.004	−0.403	0.527
rural	0.311	0.24	0.402 **	0.189	0.19	0.245
profit	0.134	0.122	0.184 *	0.095	0.076	0.13
policy	−0.094	0.067	0.214 ***	0.07	0.274 ***	0.096
cons	1.01 ***	0.197	0.927 ***	0.154	1.551 ***	0.397
F	16.7 ***		17.95 ***		13.97 **	
Observations	169		247		247	

Note: coef is the coefficient of each variable; SE is the standard deviation; cons is a constant term; ***, **, * indicate significance at the 1%, 5%, and 10% significance level, respectively. Source: Uncomtrade, “China Dairy Industry Yearbook”, “Compilation of National Agricultural Product Production Income Data”.

**Table 4 ijerph-19-02911-t004:** Influence of technology content of import source country on technological progress of raw milk production.

Country	Techch				Techch		
Import Source Country	Coef	SE	*p*	Import Source Country	Coef	SE	*p*
New Zealand	1.248	0.127	0.000	Australia	0.415	0.095	0.000
USA	−1.122	0.157	0.000	Canada	−0.123	0.043	0.005
Uruguay	−0.033	0.027	0.220	France	/	/	/
German	0.427	0.111	0.000	UK	−0.081	0.031	0.009
Denmark	−0.451	0.102	0.000	The Netherlands	0.944	0.139	0.000
Finland	−0.639	0.125	0.000	Belgium	−0.328	0.052	0.000
Irish	/	/	/	Poland	/	/	/
city	−0.0001	0.002	0.951	rural	0.079	0.123	0.518
profit	0.077	0.061	0.208	policy	/	/	/
cons	−2.01	0.758	0.000				

Note: coef is the coefficient, SE is the standard deviation, *p* is the significance, and cons is the constant term.

## Data Availability

The data presented in this study are available on request from the corresponding author.

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
