# Peer review of "Impact of Dairy Imports on Raw Milk Production Technology Progress in China"

_ijerph, 2022, doi:10.3390/ijerph19052911_

Round 1

Reviewer 1 Report

The efforts taken by the authors to model the technological progress of dairy product imports of raw milk production in China is great research work. And this model system is useful to readers, researchers and industries to predict the growth of technology. This is a great research work done by the authors.

  • Kindly explain the novelty of this research and whether this model can be used or applied to any other food commodities or anything other than raw milk to any dairy products
  • Line 263 -268 tells about data sources, kindly add reference to the obtained data sources. 
  • Kindy explain the methodology to determine the significant difference. 
  • However, one thing that the authors can mention in conclusion is how this model can be extended to other food commodities.  Conclusions can be more summarized with future research directions. 

Author Response

Thank you for spending valuable time providing quite useful comments on our work. Your comments have helped us to further improve the manuscript. We have carefully revised the manuscript according to reviewer’s suggestions. Please find our revisions as follows:  

Comment 1:Kindly explain the novelty of this research and whether this model can be used or applied to any other food commodities or anything other than raw milk to any dairy products.

Response: The novelty of this paper is that the existing research by Chinese scholars on the impact of imported dairy products on the domestic dairy industry has only focused on the “quantity” , ignoring the impact of the technological content contained in imported dairy products on the technological progress of the domestic dairy industry;The "Sanlu" milk powder incident caused Chinese consumers to lack confidence in the safety of the domestic dairy industry, while developed countries in the dairy industry such as New Zealand, Australia, the European Union, and the United States have advantages in cow hybridization technology, breeding technology, pasture management, and processing technology,the above regions are China's main source of dairy imports, Modern international trade theory believes that the division of labor between countries exists not only at the level of different products, but also at the level of different qualities of the similar product. Therefore, this article will take the “quality” of imported dairy products as the starting point, and use the technological content contained in imported dairy products to represent the quality of imported dairy products, and then evaluate the impact of the technological content of imported dairy products on the technological progress of raw milk production in China from the perspective of “quality”. It provides a reference for promoting the improvement of China's raw milk production technology, promoting China's transformation from a large dairy country to a strong dairy country, and realizing the goal of revitalizing China's dairy industry.

  If China continues to import a certain commodity in large quantities for a long time, then the imported commodity has the competitive advantage of high technological content and low cost in China, forming a substitute for Chinese related commodities. For example, due to lower price and higher quality factors, China continues to import a large amount of corn, soybeans, wool and other agricultural products every year,so this research model can be used to study the impact of the technological content contained in other imported agricultural products, livestock products, and industrial products on the technological progress of related commodities in China.

Comment 2: Line 263 -268 tells about data sources, kindly add reference to the obtained data sources.

Response: The data used in tline 263-268 comes from UNcomtrade.And I have added data sources in Word version.

Comment 3:Kindy explain the methodology to determine the significant difference

Response: In this paper, the technological added value is used to represent the technological content of imported dairy products; the technological progress index is used to represent the technical progress of raw milk production in China. Section 3.1 introduces how to calculate the technological added value that contained in imported dairy products, and Section 3.2 introduces how to use TFP to obtain the technological progress index of raw milk production in China.

  Conclusions 1 indicates that “There are significant differences in the technological content of dairy products imported from the main source countries”. The reason why different source countries have different technological content is that: According to Theory of Comparative Advantage,dairy products produced in high-income countries have high technological content, while dairy products produced in low-income countries have low technologcial content. Meanwhile, in economics, technical level is usually replaced by productivity, and labor productivity is usually used to replace productivity, that is, the added value created by each worker;that is, GDP per capita without considering demographic structure. There is a positive relationship between raw milk production technology level and per capita GDP, and the per capita GDP index is helpful to compare the development level of dairy productivity in big dairy countries. As we know that different source countries also have its own “the total export value of goods,the total export value of dairy products, the per capita GDP, and the total value of dairy products exported to China, the total value of various types of dairy products exported to China”, then according to the steps of Equation (1), (2), (3),(4), i calculated different source countries’ own technological content. So “there are significant differences in the technological content of dairy products imported from the main source countries”.

Conclusion 4 indicates that “According to the country classification, the impact of imported dairy product technological content on China's moderate scale production technology progress index of raw milk is calculated, and there are significant differences in the results” , the reason of conlusion 4 is similar to conclusion 1’s reason, because different source countries have its own technological content, so they have its own impact on China’s moderate scale technology progress index and each country has a different impact impact on China.

Comment 4: However, one thing that the authors can mention in conclusion is how this model can be extended to other food commodities. Conclusions can be more summarized with future research directions.

Response: China has a large population, and its per capita arable land resources and per capita water resources are lower than the world average. At the same time, it does not have comparative advantages in agricultural breeding, crop cultivation management, cost control, and large-scale production, Therefore, China needs to import a large amount of  agricultural that with low price and high quality products every year to meet domestic demand. In actual international trade, China imports a large amount of soybeans, corn, wool and other agricultural products every year, so the research can be extended to the impact of the technological content of soybeans, corn, wool and other agricultural products on the technological progress of related agricultural products in China.

I have supplemented the research outlook after the conclusion part in Word version.

At last: Thank the referee for providing all these detailed errors and we have gone through them and revised them in the manuscript.

Reviewer 2 Report

Line 3 and 4 Title : It is better the word "healthy" in the tittle is delete, unless there is an explanation in the"introduction. What do you mean by "healthy raw milk technology progress.

line 36. It is better to write down the long word of acronym first when it is written in the text for the first time, then write the acronym.  In the nexk sentense, an acronym can write.

36 : what the long word CIF, ?

All of my comment can be see in the manucript

Author Response

Thank you for spending valuable time providing quite useful comments on our work. Your comments have helped us to further improve the manuscript. We have carefully revised the manuscript according to reviewer’s suggestions. Please find our revisions as follows:  

Comment 1:Line 3 and 4 Title : It is better the word "healthy" in the tittle is delete, unless there is an explanation in the"introduction. What do you mean by "healthy raw milk technology progress.

Response: This is a translation mistake. I’m sure that "health" is a superfluous word, it's irrelevant to the full text.The correct title of the article is: Impact of dairy imports on raw milk production technology progress in China.

Comment 2: Line 36. It is better to write down the long word of acronym first when it is written in the text for the first time, then write the acronym. In the nexk sentense, an acronym can write.

Response: Thank you for telling me how to standardize the use of acronyms in English writing. I have modified as your suggestion, and the corrected use of CIF is Cost, Insurance and Freight ( CIF ),you can see it in Word version with red color in line 36.

Comment 3: Line 36. What the long word CIF ?

Response: CIF is a special term for price settlement in international trade. Its full name is Cost, Insurance and Freight, which means that the seller is responsible for the cost of the goods, insurance premiums and shipping costs to the buyer's port.

All my response can be seen in PDF, and in Word version with red color.

At last:Thank the referee for providing all these detailed errors and we have gone through them and revised them in the manuscript.

Round 2

Reviewer 2 Report

This research is very interesting, and you have completed the information.